# Two XMAP215/TOG Microtubule Polymerases, Alp14 and Dis1, Play Non-Exchangeable, Distinct Roles in Microtubule Organisation in Fission Yeast

**DOI:** 10.3390/ijms20205108

**Published:** 2019-10-15

**Authors:** Masashi Yukawa, Tomoki Kawakami, Corinne Pinder, Takashi Toda

**Affiliations:** 1Division of Biological and Life Sciences, Graduate School of Integrated Sciences for Life, Hiroshima University, 1-3-1 Kagamiyama, Higashi-Hiroshima, Hiroshima 739-8530, Japan; myukawa@hiroshima-u.ac.jp (M.Y.); m176779@outlook.com (T.K.); corinnepinder@outlook.com (C.P.); 2Hiroshima Research Center for Healthy Aging (HiHA), Hiroshima University, 1-3-1 Kagamiyama, Higashi-Hiroshima, Hiroshima 739-8530, Japan; 3The Francis Crick Institute, 1 Midland Road, London NW1 1AT, UK

**Keywords:** fission yeast, microtubule polymerase, XMAP215/TOG, mitotic spindle, spindle pole body, kinetochore

## Abstract

Proper bipolar spindle assembly underlies accurate chromosome segregation. A cohort of microtubule-associated proteins orchestrates spindle microtubule formation in a spatiotemporally coordinated manner. Among them, the conserved XMAP215/TOG family of microtubule polymerase plays a central role in spindle assembly. In fission yeast, two XMAP215/TOG members, Alp14 and Dis1, share essential roles in cell viability; however how these two proteins functionally collaborate remains undetermined. Here we show the functional interplay and specification of Alp14 and Dis1. Creation of new mutant alleles of *alp14*, which display temperature sensitivity in the absence of Dis1, enabled us to conduct detailed analyses of a double mutant. We have found that simultaneous inactivation of Alp14 and Dis1 results in early mitotic arrest with very short, fragile spindles. Intriguingly, these cells often undergo spindle collapse, leading to a lethal “cut” phenotype. By implementing an artificial targeting system, we have shown that Alp14 and Dis1 are not functionally exchangeable and as such are not merely redundant paralogues. Interestingly, while Alp14 promotes microtubule nucleation, Dis1 does not. Our results uncover that the intrinsic specification, not the spatial regulation, between Alp14 and Dis1 underlies the collaborative actions of these two XMAP215/TOG members in mitotic progression, spindle integrity and genome stability.

## 1. Introduction

Microtubules (MTs) play diverse roles in a wide range of biological processes including cell cycle progression, development and differentiation pathways [1]. During mitosis, spindle MTs assemble to form a bipolar structure that emanates from the two spindle poles. The bipolar spindle functions as the division apparatus for sister chromatids, generating pulling forces to move them towards opposite poles to ensure equal partition of genetic material. Errors in this process can lead to cell death and/or aneuploidy, a major risk factor for miscarriage, birth defects and tumourigenesis [2,3].

MTs are intrinsically dynamic, a characteristic termed dynamic instability [4,5]. In a living cell, a cohort of proteins, collectively known as microtubule-associated proteins (MAPs), participate in spindle assembly, stability and maintenance as structural and/or regulatory factors. A conserved family of MAPs, here referred to as the XMAP215/TOG (Tumour Overexpressed Gene) family, is deeply rooted in the eukaryotic branch of the tree of life and arguably the most important regulator in MT organisation [6,7,8,9]. Originally discovered as a factor conferring the dynamic properties of MTs in *Xenopus* egg extracts [10], homologues have been identified in virtually all eukaryotic species [9]. The seminal advancement of our understanding of the XMAP215/TOG family was the discovery that XMAP215 is a MT polymerase [11]; this protein is capable of incorporating α/β-tubulin heterodimers at the plus end of pre-existing MTs. Since then, this activity has been proven for other members of the family from a diverse range of organisms [8]. Consistent with this notion, depletion or mutational inactivation of XMAP215/TOG family members generally results in the emergence of shorter, unstable MTs in various cell types [8].

Members of the XMAP215/TOG family have similar domain organisation. The N-terminal region contains multiple (two to five) TOG domains that bind and trap free α/β- tubulin heterodimers [12,13], followed by the MT lattice-binding domain and finally the C-terminal region responsible for interaction with various binding partners [8]. Fission yeast is unique in that its genome contains two XMAP215/TOG paralogues, *alp14*/*mtc1* and *dis1* [14,15], unlike other organisms that normally encode only one gene [9]. This provides an opportunity in which to study the conservation and diversification of the XMAP215/TOG family members within the same organism. It has been shown that both Alp14 and Dis1 behave biochemically as canonical MT polymerases [16,17]. Genetically, *alp14* and *dis1* share essential functions for cell viability; while each single deletion is viable, strains containing double gene deletions are inviable [18]. These phenomena imply that Alp14 and Dis1 are redundant paralogues. However, several lines of data are not consistent with this simple scenario. For instance, Alp14 and Dis1 form a complex with different MAP partners in a cell; Alp14 binds Alp7/TACC (Transforming Acidic Coiled-Coil), while Dis1 interacts with Mal3/EB1 [16,19]. In addition, during mitosis, the cellular localisation of Alp14 and Dis1 is different; Alp14 is first recruited to the spindle pole body (SPB, the fungal equivalent of the centrosome), while Dis1 is localised to the kinetochore [15,20,21,22,23].

Interestingly, it has been shown that Alp14 is capable of promoting the very initial step of MT assembly at the SPB, namely MT nucleation by binding to the universal MT nucleator γ-tubulin complex (γ-TuC) [24], which is also the case for XMAP215 [25]. Whether or not Dis1 could also promote MT nucleation remains to be determined. By contrast, we and others have recently shown that Dis1 is capable of promoting MT depolymerisation, rather than polymerisation, under some physiological circumstances [26,27]. 

In this study, we have addressed the reason for synthetic lethality between *alp14* and *dis1* by creating and characterising new *alp14* mutant alleles that display temperature sensitivity in the absence of Dis1. Our analyses show that Alp14 and Dis1 collaborate in assembly and maintenance of bipolar spindle MTs as MT polymerases. Intriguingly, we have found that Alp14 and Dis1 are not redundant paralogues, but instead these MAPs play intrinsically diversified roles in spindle MT assembly in a spatially distinct manner.

## 2. Results

### 2.1. Creation of Temperature-Sensitive alp14 Mutants in the Absence of Dis1

To explore the physiological and phenotypic consequences resulting from simultaneous inactivation of two XMAP215/TOG paralogues in fission yeast, we sought to isolate temperature-sensitive (ts) mutants of *alp14* in the *dis1* deletion (*dis1∆***)** background. For this purpose, a pool of *alp14-GFP-kanR* fragments amplified with error-prone PCR [28] was introduced into a *dis1∆* strain and G418-resistant colonies were first selected at 27 °C. These colonies were replica-plated and incubated at 36 °C. From this screen, we isolated 6 ts candidates (Materials and Methods). These ts isolates were then backcrossed with a wild-type strain and random spore analysis was performed to check whether temperature-sensitivity cosegregated with G418 resistance (*alp14-GFP-kanR*) and Hygromycin B resistance (*dis1::hphR*). Spot tests were performed to assess the ts phenotype of individual mutants (Figure 1A). After confirming these characteristics, the nucleotide sequence of the *alp14* gene in individual mutants was determined. Each mutant contained between one and five point mutations and mutation sites were spread across the entirety of Alp14. This includes the N-terminal TOG repeats, the C-terminal MT-binding region and the last 100 amino acid sequence that is required for binding to Alp7/TACC [19] (Figure 1B). We chose *alp14-26* for subsequent analyses as this mutant allele displayed strong temperature sensitivity in the *dis1∆* background (no growth at temperatures above 34 °C) and contained only a single point mutation (C450R) (Figure 1A,B, respectively).

### 2.2. Alp14 and Dis1 Collaborate to Regulate Proper Mitotic Progression and Promote Spindle Elongation

As the first step towards characterising the functional interplay between Alp14 and Dis1, we assessed the mitotic behaviour of *alp14-26dis1∆* mutant cells incubated at 36 °C for 2 h and compared them with those of individual single mutants, *alp14∆* and wild-type cells. In either *dis1∆* or *alp14-26* single mutant cells the mitotic index, judged by the presence of spindle MT structures (mCherry-Atb2) [29], was increased compared to wild-type cells (23% and 38% respectively vs 15% in wild-type cells, Figure 2A). In the *alp14∆* strain, the mitotic index was higher (47%), indicating that *alp14-26* is hypomorphic and may retain some residual activity. Interestingly, *alp14-26dis1∆* cells exhibited a mitotic index of 60%, higher than that in *alp14∆*, indicating that Alp14 and Dis1 regulate mitotic progression in an independent yet additive fashion (Figure 2A).

Mitosis in fission yeast is divided into three distinct phases based upon spindle behaviour [30]. Phase 1 (corresponding to prophase and prometaphase) constitutes the period in which spindles start to assemble and elongate until a fixed length of ~2.5 µm; in phase 2 (corresponding to metaphase and anaphase A) the spindle is maintained at a constant length of ~2.5 µm; phase 3 (corresponding to anaphase B) involves final spindle elongation from ~2.5 to ~10 µm. We measured spindle length in each strain and found that although *dis1∆* cells did not display a significant difference from wild-type cells (3.4 µm vs 3.8 µm, respectively), *alp14-26* cells showed a reduction (2.3 µm), yet were longer than *alp14∆* cells (1.5 µm). Consistent with the additive impact of these two MAPs on mitotic index, *alp14-26dis1∆* cells showed the shortest length (0.7 µm) (Figure 2B), suggesting that mutant cells stay in this stage for a longer period of time. Thus, Alp14 in collaboration with Dis1 plays an important role in proper spindle length control.

### 2.3. Alp14, But Not Dis1, Promotes Microtubule Nucleation

It is known that in fission yeast, as in other species, the homologue of polo-like kinase Plo1 is localised to the SPB from the end of G2 phase until anaphase; fluorescently tagged Plo1 serves as a reliable marker for early to mid-mitosis [31,32,33]. Observation of Plo1-GFP at SPBs in various strains reinforced our previous results, showing collaborative action between Alp14 and Dis1 (see Figure 2A,B). The percentage of cells possessing Plo1-GFP signals on the two SPBs was increased in *dis1∆* or *alp14∆* compared to wild-type cells (18% or 24% respectively vs 8% for wild-type cells) (Figure 2C). Although the percentage of these Plo1-GFP positive cells was not significantly altered (10%) in the *alp14-26* mutant, *alp14-26dis1∆* cells showed a substantial increase (35%) (Figure 2C). This result supports the notion that Alp14 and Dis1 collaborate to promote spindle elongation.

Interestingly, we found that the percentage of cells displaying a single Plo1-GFP signal on the unseparated SPB was the same between wild-type and *dis1∆* cells (<1%, n.s.), and further, the percentage did not increase significantly in *alp14-26dis1∆* cells compared to *alp14-26* cells (15-16%, n.s.) (Figure 2D). As most of these cells (85%, 35/41) did not contain interphase MTs but instead dot-like MT signals were observed around the SPB (Figure 2E), they were posited to be in an early mitotic stage. Note that the intensity of these dots appeared to be severely reduced in *alp14-26* compared to wild-type cells by visual inspection, consistent with the notion that Alp14 is required for MT nucleation [17,24]. Thus, Alp14 and Dis1 are required for proper spindle elongation; however only Alp14 promotes MT nucleation.

### 2.4. Microtubule Intensities Are Reduced by Simultaneous Inactivation of Alp14 and Dis1

The data discussed above indicate *alp14-26dis1∆* cells spend a longer period of time during an early mitotic stage with shorter spindle length. To study the defects associated with this delay, we measured the intensity of spindle MTs in individual cells. We found that spindle intensity was markedly reduced in the *alp14-26dis1∆* mutant (~50 % reduction compared to wild-type cells, Figure 3A,B). In either of *alp14-26* or *alp14∆* cells, spindle intensities were also decreased compared to wild-type cells (~30% reduction), but higher than those in the *alp14-26dis1∆* mutant (Figure 3A,B). This result corroborates the notion that Alp14 and Dis1 are together yet independently required for spindle assembly and elongation. It is worth noting that as spindle intensity of *dis1∆* cells did not display an obvious reduction, Alp14 appears to play a compensatory role in the maintenance of spindle intensity in the absence of Dis1, which is also the case for spindle length control (see Figure 2B).

Previous work found that Alp14 is required for MT nucleation as well as MT growth during interphase [17,24]. By contrast, Dis1 is involved in the stabilisation of MT bundles during interphase [34]. Given these previous findings, we were interested in the structures of interphase MTs in *alp14-26dis1∆* double mutants. Visual inspection showed that *alp14-26dis1∆* mutant cells exhibit remarkably faint interphase MT structures; these MTs were much shorter and dimmer compared to wild-type and each single mutant (Figure 3C). Therefore, Alp14 and Dis1 in concert promote growth and stability of microtubules during both interphase and mitosis.

### 2.5. Short Spindles Often Collapse in alp14-26dis1∆ Cells, Leading to a Lethal “Cut” Phenotype

We next investigated the behaviour of live spindle MTs by time-lapse microscopy. This analysis uncovered that many mitotic *alp14-26dis1∆* cells failed to elongate spindles properly; in 78% (21/27) of mitotic cells, the two SPBs remained in close proximity for a prolonged period of time (Figure 4A,B). This result mirrored those obtained from still images in our initial experiments (see Figure 2B,C). Close inspection of live images revealed that in these cells the SPBs exhibited a “breathing” or “’collapse” behaviour: the distance between the two SPBs fluctuated (Figure 4C, top) or was shortened substantially (Figure 4C, bottom), respectively. In wild-type cells, we have never observed this defect (*n* = 20). Hence, Alp14 and Dis1 are required for the maintenance of spindle MT integrity.

Additionally, we found that *alp14-26dis1∆* exhibited a lethal “cut” phenotype (Figure 4D), in which septation occurred in the absence of proper chromosome segregation [35]. Although we have not observed the cut phenotype in wild-type cells (*n* = 35), all the single mutants displayed this defect to varying degrees, though the frequency was much lower compared to *alp14-26dis1∆* double mutant cells (6–19% vs 31%, respectively) (Figure 4E). Thus, the reason for lethality of *alp14-26dis1∆* cells incubated at the restrictive temperature is attributable to untimely septation in the absence of proper sister chromatid segregation.

### 2.6. Alp14 and Dis1 Are Not Functionally Exchangeable

Our results thus far indicate that the two fission yeast XMAP215/TOG paralogues collaborate to ensure the growth and stability of spindle MTs. Previously, we showed that overproduction of *alp14*^+^ is capable of rescuing the cold sensitivity of *dis1∆*, and overproduction of *dis1*^+^ suppresses the ts defect of *alp14∆* [14]. In addition, *dis1∆alp14∆* double deletion is synthetically lethal at temperatures at which single deletion strains are viable [18]. These results suggest that Alp14 and Dis1 are functionally exchangeable paralogues; therefore, we were interested in understanding if these two proteins are replaceable at the endogenous level (without overproduction).

The most notable difference between Alp14 and Dis1 is their cellular localisation during mitosis. Alp14 is first localised to the SPB, followed by translocation to the mitotic spindle and movement towards the kinetochore as spindles elongate and capture it [36]. Dis1 is localised to the kinetochore first and loaded onto the mitotic spindle only at a later mitotic stage; importantly, Dis1 is not localised to the SPB [16,20,37]. One possible scenario is that once Dis1 is recruited to the SPB, it may be possible to functionally replace Alp14. To address this notion, we implemented the GFP-GFP-binding protein (GBP) protein targeting system [38]. To this end, *alp7∆*, *alp14∆* and wild-type strains were constructed that carried Dis1-GFP and GBP-Alp4 (Figure 5A); Alp4/GCP2 is a core component of the γ-TuC that is constitutively localised to the SPB [39]. To visualise GBP-Alp4, it was also tagged with the fluorescent protein mCherry (GBP-mCherry-Alp4). Tethering Dis1 to the SPB was successful in all strains (Figure 5B). However, assessment of the growth of these strains indicated that forced recruitment of Dis1 to the SPB was not sufficient in the absence of Alp7 or Alp14; these strains remained ts like *alp7∆* or *alp14∆* strains containing only Dis1-GFP in the absence of GBP-mCherry-Alp4 (Figure 5C). It is of note that Dis1-GFP in the presence of GBP-mCherry-Alp4 was functional, as the corresponding wild-type construct normally formed colonies at either 22 °C or 36 °C (Figure 5C). T hus, a simple recruitment of Dis1 to the SPB is not sufficient to replace the functionality of Alp14.

Next, the converse experiment was performed; we tethered Alp14 to the kinetochore in a *dis1∆* strain. To do so, we exploited Mis12-GBP-mCherry (Figure 6A), a constitutive component of the kinetochore [40]. Alp14-GFP was localised to the kinetochore during mitosis in all strains (Figure 6B). Growth assays revealed that Alp14-GFP and Mis12-GBP-mCherry could not alleviate cold sensitivity of *dis1∆* cells (Figure 6C), suggesting that kinetochore-tethered Alp14 cannot compensate for the loss of Dis1. Alp14-GFP in the presence of Mis12-GBP-mCherry is functional, as wild-type cells containing these two fluorescent proteins could form normal colonies at all temperatures assessed (Figure 6C). Taken together, the results shown here indicate that the spatial difference does not account for the functional specification between Alp14 and Dis1. Therefore, we conclude that although Alp14 and Dis1 are structurally related MT polymerases and act in a collaborative fashion, these two MAPs are functionally divergent and execute their individual roles in a spatially distinct manner.

## 3. Discussion

In this study, we have explored the functional conservation and diversification of two fission yeast XMAP215/TOG paralogues Alp14 and Dis1. We show that simultaneous inactivation of Alp14 and Dis1 has an adverse impact on spindle assembly in an additive fashion; double mutant cells arrest in early mitosis with very short, fragile spindles. These cells often undergo spindle collapse, leading to a lethal cut phenotype. Intriguingly, mutual spatial replacement between these two MAPs is unable to complement the function for the other. Therefore, in opposition to a simple scenario, our results have illuminated the intrinsic functional specification between Alp14 and Dis1 (Figure 7).

### 3.1. Alp14 and Dis1 Have Evolved to Execute Functionally Non-Exchangeable Roles

In normal cells, Alp14 is recruited to the mitotic SPB in the nucleus where it promotes spindle assembly [19,43,44,45,46]. It is known that Alp7 mediates this translocation [47,48] and Alp7 function can be bypassed through direct targeting of Alp14 to the nuclear SPB [42]. Importantly, we show here that the analogous tethering for Dis1 is not sufficient to circumvent Alp14 function. Conversely, tethering Alp14 to the kinetochore, the critical structure at which Dis1 plays a role in stabilisation of kinetochore-spindle attachment [20], is incapable of bypassing the need for Dis1. Together, these results have established that Alp14 and Dis1 are functionally non-exchangeable. This conclusion is in line with phylogenetic analyses, in which Alp14 and Dis1 split relatively early during fungi diversification: these two genes are diversified before the emergence of various fungi species yet after fungi separate from animals [9]. It is worth mentioning that unlike Alp14 and Dis1 MT polymerases, two kinesin-14 members in fission yeast, Pkl1 and Klp2, are functionally replaceable by artificial targeting of Klp2 to the SPB to which Pkl1, but not Klp2, is normally localised [49].

An interesting question is then how does a single member of the XMAP215/TOG family in other organisms perform the equivalent roles of Alp14 and Dis1? It is known that in animals including humans, a stable interaction between XMAP215/TOG members and TACC homologues is widely conserved like Alp14 and Alp7/TACC [8,50]. Intriguingly, in metazoans, multiple TACC homologues exist, by which individual TACC members might execute specific roles in MT organisation by forming distinct complexes with XMAP215/TOG [50]. In addition, it is known that animal XMAP215/TOG interacts with EB1 through adaptor proteins: Sentin in flies [51] and SLAIN in humans [52], and XMAP215 and EB1 synergistically increase MT growth rate at the plus end [53]. This may underpin the role of animal XMAP215/TOG in the stabilisation of proper kinetochore-spindle attachment, like the Dis1-Mal3 complex [16]. As aforementioned, Alp14 and Dis1 are evolutionarily diverged only after the fungi kingdom separated from the animal kingdom [9]. Hence, animal XMAP215/TOG members, like their ancestor, may retain dual activities that Alp14 and Dis1 each execute. Alternatively, albeit not mutually exclusive, Dis1 and/or Alp14 may acquire unique roles specific for fission yeast.

### 3.2. Dis1 May Be a Specialised Regulator of Microtubule Dynamics at the Kinetochore-Microtubule Interface

It is known that metazoan XMAP215/TOG, budding yeast Stu2 and fission yeast Alp14 are critical factors for MT nucleation, in addition to MT polymerisation, at the centrosome/SPB [24,25,54,55]. As for Dis1, this protein is neither recruited to the SPB [15,20] nor capable of replacing the role of Alp14 even if it is targeted to this site. Accordingly, we consider that unlike other members of the XMAP215/TOG family, Dis1 has lost its MT nucleating activity through evolution. The fact that in the *alp14-26dis1∆* double mutant very early mitotic cells accumulate indicates that Dis1 plays a role in the stabilisation of spindle MTs from this early mitotic stage. 

Previous studies have highlighted unexpected, unique functions of Dis1 compared to other XMAP215/TOG members. For instance, during the first meiotic division, Dis1 plays a role in microtubule shortening to pull kinetochores polewards: this protein appears to promote MT depolymerisation under this condition [26]. In addition, we recently showed that Dis1 potentiates spindle shortening in the absence of the Klp5-Klp6 kinesin-8 complex to ensure kinetochore capture by spindle MTs for proper mitotic progression [27]. Furthermore, Dis1, but not Alp14, plays a role in epigenetic regulation at centromeric chromatin [56]. Overall, Dis1 appears committed to a specific task at the kinetochore-MT interface during mitosis, thereby stabilising proper attachment [20]. Identification of the domains and amino acid residues within Alp14 and Dis1 that delimit Alp14- or Dis1-specific activities would be of great significance to delineate how individual members of the XMAP215/TOG family execute conserved and diversified roles and also to obtain mechanistic information regarding how this family promotes MT polymerisation and/or depolymerisation.

## 4. Materials and Methods

### 4.1. Strains, Media, and Genetic Methods

Fission yeast strains used in this study are listed in Table 1. Media, growth conditions and manipulations were performed as previously described [57]. For most of the experiments, rich YE5S liquid media and agar plates were used. Spot tests were performed by spotting 5–10 μL of cells at a concentration of 2 × 10^7^ cells/mL after 10-fold serial dilutions onto rich YE5S plates. Some of the YE5S plates also contained Phloxine B, a dye that accumulates in dead or dying cells and stains those colonies dark pink due to a reduced ability to pump out the dye. Plates were incubated at various temperatures from 22 °C to 36 °C as necessary.

### 4.2. Preparation and Manipulation of Nucleic Acids

Enzymes were used as recommended by the suppliers (New England Biolabs Inc. Ipswich, MA and Takara Bio Inc., Shiga, Japan).

### 4.3. Strain Construction, Gene Disruption and N-Terminal and C-Terminal Epitope Tagging

A PCR-based gene-targeting method [58,59] was used for complete gene disruption and epitope tagging (e.g., GFP and mCherry) at the C-terminus, by which all the tagged proteins were produced under the endogenous promoter. Construction of strains containing Alp14^NLS^-GFP or GBP-mCherry-Alp4 was described previously [42].

### 4.4. Isolation of alp14 Temperature-Sensitive Mutants in the dis1∆ Background

The *alp14* ts mutants were constructed by PCR-based random mutagenesis as previously reported [28,42]. A *GFP* and G418-resistance marker gene cassette (*kanR*) was inserted in-frame prior to the stop codon of the *alp14* gene (*alp14-GFP-kanR*). The *alp14-GFP-kanR* fragment extracted and purified from this strain was amplified with error-prone PCR using *TaKaRa EX taq* polymerase (Takara Bio Inc. Shiga, Japan). After ethanol precipitation, pooled amplified fragments were introduced into a *dis1∆* (*dis1::hphR*) strain and G418-resistance transformants were selected at 27 °C. We obtained 6 ts isolates from ~5000 G418 resistant colonies. These mutants were crossed with a wild-type strain and random spore analysis was performed. In all segregants, the ts phenotype co-segregated with G418- and Hygromycin B-resistance. Subsequently, nucleotide sequencing was performed to determine the mutation sites in the *alp14* gene of these mutants.

### 4.5. Fluorescence Microscopy and Time-Lapse Live Cell Imaging

Fluorescence microscopy images were obtained using a DeltaVision microscope system (DeltaVision Elite; GE Healthcare, Chicago, IL, USA) with a wide-field inverted epifluorescence microscope (IX71; Olympus, Tokyo, Japan) and a Plan Apochromat 60×, NA 1.42, oil immersion objective (PLAPON 60×O; Olympus Tokyo, Japan). DeltaVision image acquisition software (softWoRx 6.5.2; GE Healthcare, Chicago, IL, USA) equipped with a charge-coupled device camera (CoolSNAP HQ2; Photometrics, Tucson, AZ, USA) was used. Live cells were imaged in a glass-bottomed culture dish (MatTek Corporation, Ashland, MA, USA) coated with soybean lectin and incubated at 27 °C or at 36 °C. The latter were cultured in rich YE5S media until mid–log phase at 27 °C and subsequently shifted to the restrictive temperature of 36 °C for 2 h before microscopic observation. To maintain cells at the proper temperature during imaging, a temperature-controlled chamber (Air Therm SMT; World Precision Instruments Inc., Sarasota, FL, USA) was used. Time-lapse imaging was performed for 30 min at 36 °C. Images were taken as 14–16 sections along the z-axis at 0.2 μm intervals. The sections of images acquired at each time point were compressed into a 2D projection using the DeltaVision maximum intensity algorithm. Deconvolution was applied before the 2D projection. Captured images were processed with Photoshop CS6 (version 13.0; Adobe, San Jose, CA, USA).

### 4.6. Quantification of Fluorescent Signal Intensities

For quantification of signal intensities of the fluorescent marker-tagged protein (e.g., mCherry-Atb2), 14–16 sections were taken along the z-axis at 0.2 μm intervals. Projection images of maximum intensity were obtained after deconvolution, and upon subtracting background intensities only values of maximum fluorescence intensities covering the whole GFP signals within the z-sections were used for statistical data analysis.

### 4.7. Statistical Data Analysis

We used the two-tailed unpaired Student’s *t*-test to evaluate the significance of differences in different strains, unless otherwise stated. All experiments were performed at least twice. Experiment sample numbers used for statistical testing were given in the corresponding figures and/or legends. We used this key for asterisk placeholders to indicate *p*-values in the figures: e.g., ****, *p* < 0.0001.

## Figures and Tables

**Figure 1 ijms-20-05108-f001:**
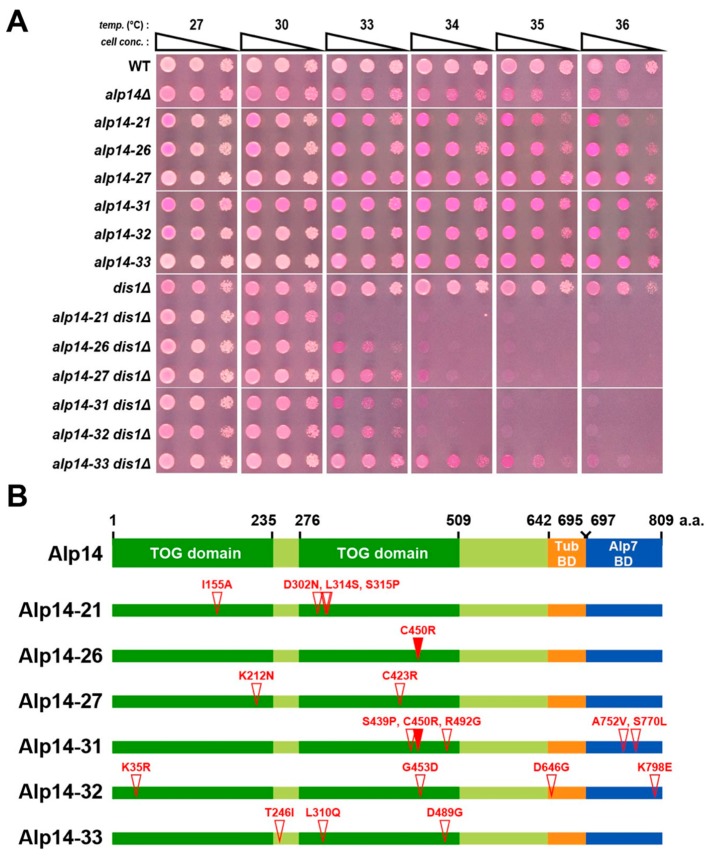
Isolation of *alp14* temperature-sensitive mutations in the *dis1∆* background. (**A**) Spot test. Indicated strains were serially (10-fold) diluted, spotted onto rich YE5S plates containing Phloxine B and incubated at the indicated temperatures for 2–3 d. (**B**) Mutation sites in the *alp14* mutants. Mutated amino acid residues are shown with open triangles. Note that the C450R mutation was found in both *alp14-26* and *alp14-31* (shown with closed triangles).

**Figure 2 ijms-20-05108-f002:**
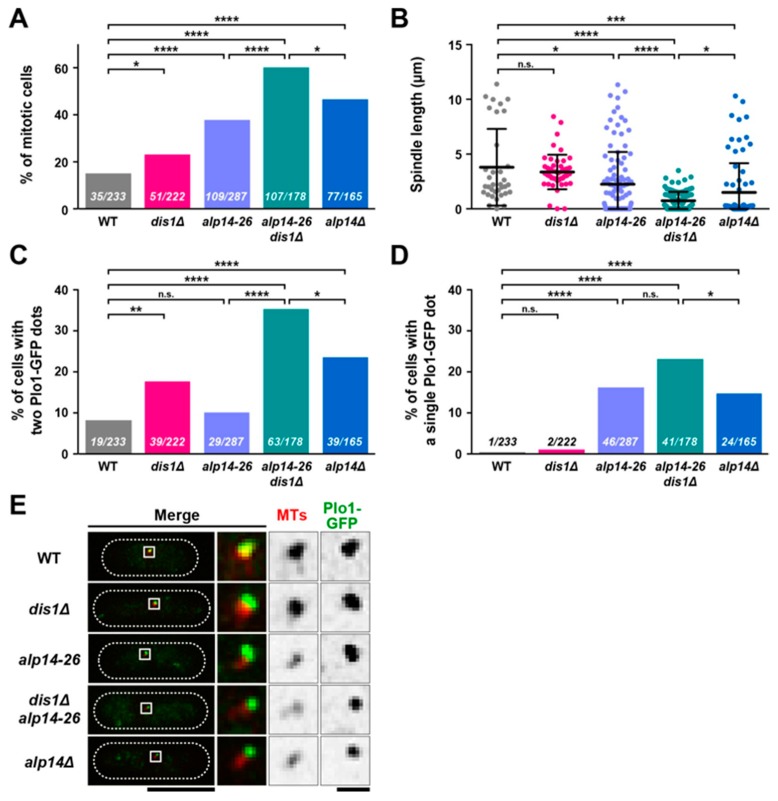
Alp14 and Dis1 collaborate to regulate proper mitotic progression. Exponentially growing cells at 27 °C were shifted to 36 °C for 2 h. Spindle MTs (mCherry-Atb2, red) and mitotic SPBs (Plo1-GFP, green) were visualised. (**A**) Mitotic indices. Mitotic cells were identified as those containing mitotic spindles or those without interphase MTs. *p*-values are derived from a two-tailed Chi-squared test (* *p* < 0.05; **** *p* < 0.0001). (**B**) Spindle length. Data sets were compared with a two-tailed unpaired Student’s t-tests (* *p* < 0.05; *** *p* < 0.001; **** *p* < 0.0001; n.s., not significant). (**C**) The percentage of cells containing Plo1-GFP signals on two SPBs. *p*-values are derived from the two-tailed Chi-squared test (* *p* < 0.05; ** *p* < 0.01; **** *p* < 0.0001; n.s., not significant). (**D**) The percentage of cells containing one Plo1-GFP signal on the unseparated SPB. *p*-values are derived from the two-tailed Chi-squared test (* *p* < 0.05; **** *p* < 0.0001; n.s., not significant). (**E**) Representative images of cells containing Plo1-GFP signals on the SPB. Cell peripheries are indicated by dotted lines and areas containing SPBs (squares) are enlarged in the three panels on the right-hand side. Scale bars, 10 μm (left) and 1 μm (right).

**Figure 3 ijms-20-05108-f003:**
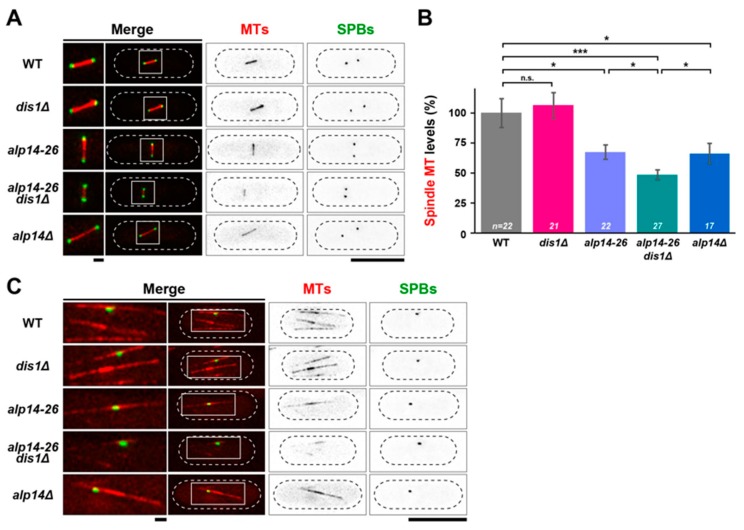
Microtubule intensities are reduced in *alp14-26dis1∆* in an additive fashion compared to each single mutant. Exponentially growing cells at 27 °C were shifted to 36 °C for 2 h. Spindle MTs (mCherry-Atb2; red) and SPBs (Cut12-GFP; green) were visualised. (**A**) Representative images of mitotic spindles. Cell peripheries are indicated by dotted lines and areas containing spindle MTs (squares) are enlarged in the left-hand side panels. (**B**) Quantification of spindle MT intensities. Fluorescence intensities of mCherry-Atb2 were measured in the indicated strains. The values of wild-type cells were taken as 100% and used for comparison to other strains. All *p*-values were obtained from a two-tailed unpaired Student’s t test. Data are presented as the means ± SEM (≥17 cells). (* *p* < 0.05; *** *p*< 0.001; n.s., not significant.) (**C**) Representative images of interphase MTs. Images were taken from samples incubated at 27 °C. Cell peripheries are indicated by dotted lines and areas containing interphase MTs (squares) are enlarged in the left-hand side panels. Scale bars, 1 μm (left) and 10 μm (right).

**Figure 4 ijms-20-05108-f004:**
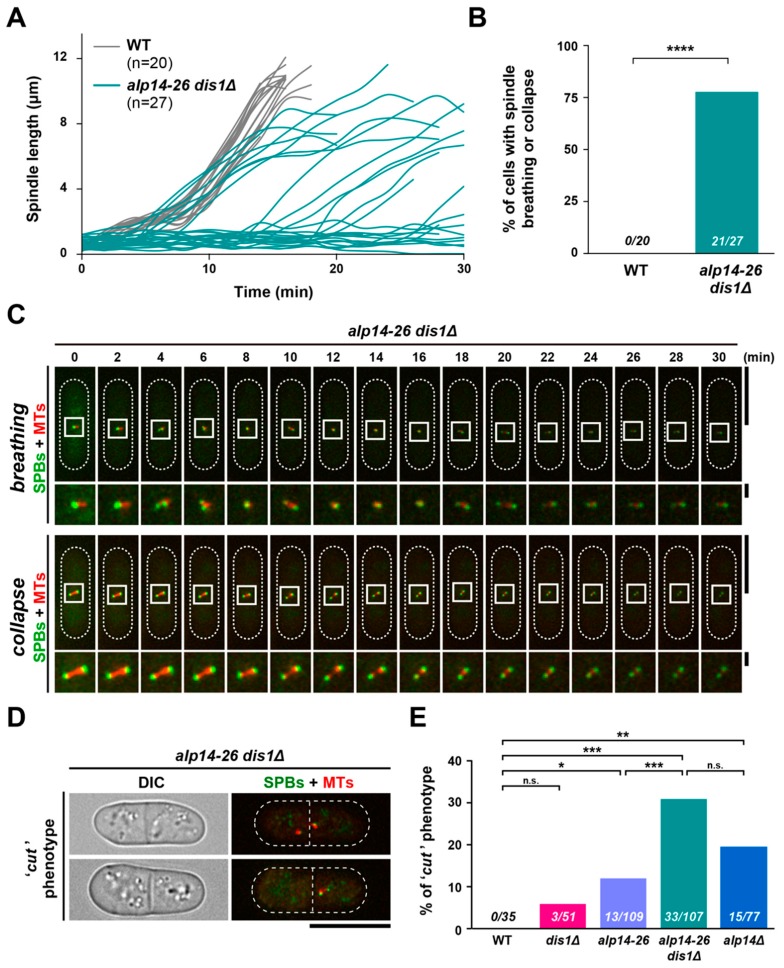
Short spindles are often collapsed in *alp14-26dis1∆*, leading to the lethal “cut” phenotype. Exponentially growing cells at 27 °C were shifted to 36 °C for 2 h. Spindle MTs (mCherry-Atb2; red) and SPBs (Cut12-GFP (**A**–**C**) or Plo1-GFP (**D** and **E**); green) were visualised. (**A**) Plots of spindle length over time. Spindle length (the distance between SPBs) obtained from live cell imaging was measured over time in wild-type (grey, *n* = 20) and *alp14-26∆dis1∆* cells (teal, *n* = 27). (**B**) Percentage of cells displaying spindle breathing or collapse. The sample numbers are shown (*n* = 20 for wild-type and *n* = 27 for *alp14-26∆dis1∆* cells). As the distinction of the spindle behaviour between breathing and collapse is sometimes equivocal, these two phenotypes are combined in this graph. The *p*-value is derived from a two-tailed Chi-squared test (**** *p* < 0.0001). (**C**) Time-lapse images of a mitotic *alp14-26dis1∆* cell displaying spindle breathing or collapse. Representative images of breathing (top) or collapse (bottom) are shown. Images were taken at 2 min intervals after 2 h incubation at 36 °C. Cell peripheries are indicated with dotted lines (the top part of each row). The areas containing SPBs and spindle microtubules are marked by square boxes and are enlarged (the bottom part of each row). Scale bars, 10 μm (top) and 1 μm (bottom). (**D**) Representative images of *alp14-26∆dis1∆* cells exhibiting the “cut” phenotype. Scale bar, 10 μm. (**E**) The percentage of cells displaying the “cut” phenotype. All *p*-values are derived from a two-tailed Chi-squared test (* *p* < 0.05; ** *p* < 0.01; *** *p* < 0.001; n.s., not significant).

**Figure 5 ijms-20-05108-f005:**
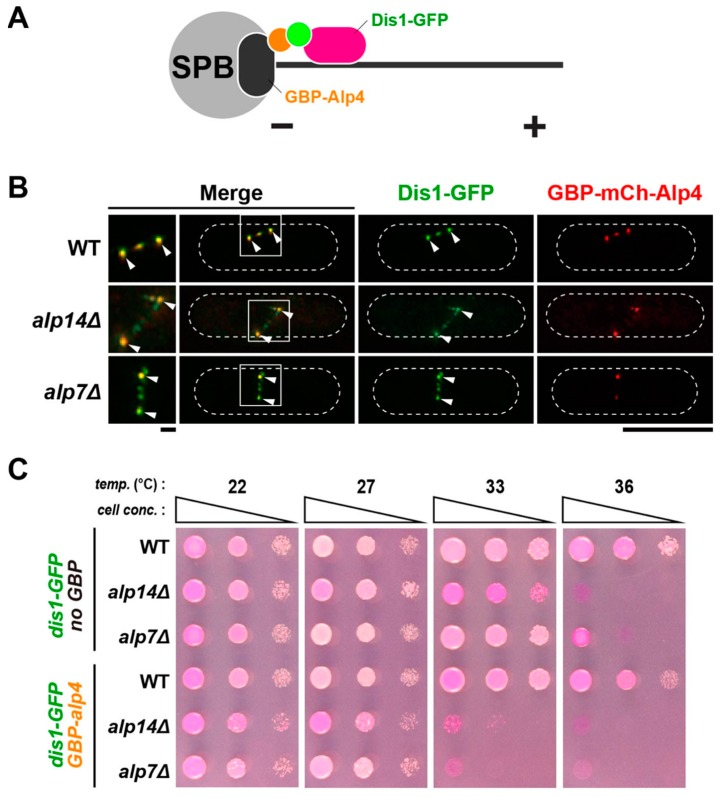
SPB-tethered Dis1 is incapable of rescuing *alp14∆*. (**A**) A schematic illustration of a strategy for tethering Dis1 to the SPB. (**B**) Visualisation of Dis1-GFP localisation in strains containing GBP-mCherry-Alp4. Cell peripheries are indicated with dotted lines and areas containing spindle MTs (squares) are enlarged in the left-hand side panels. Note that in addition to SPB localisation (arrowheads), some fractions of Dis1-GFP are localised to additional dots between the two SPBs. This location is most likely to be the kinetochore, to which Dis1 is normally localised during mitosis [15,20,37]. Scale bars, 1 μm (left) and 10 μm (right). (**C**) Spot test. Indicated strains were serially (10-fold) diluted, spotted onto rich YE5S plates containing Phloxine B and incubated at the indicated temperatures for 2-3 d.

**Figure 6 ijms-20-05108-f006:**
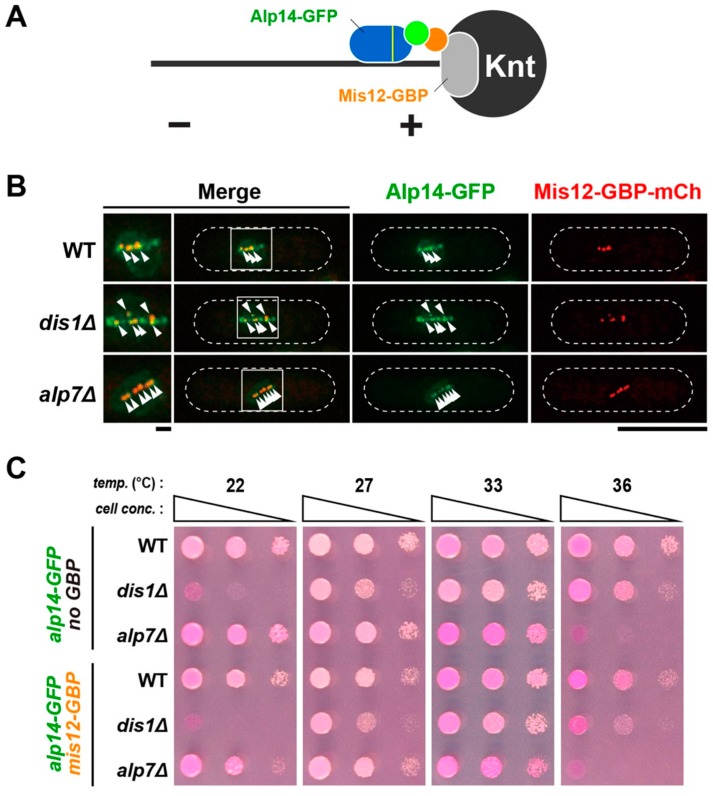
Kinetochore-tethered Alp14 is incapable of rescuing *dis1∆*. (**A**) A schematic illustration of a strategy for tethering Alp14 to the kinetochore (Knt). To ensure nuclear entry of Alp14, a canonical nuclear localisation signal (NLS) sequence [41] was added together with GFP to the C terminus of Alp14 (Alp14^NLS^-GFP) [42]. (**B**) Visualisation of Alp14^NLS^-GFP localisation in strains containing Mis12-GBP-mCherry. Cell peripheries are indicated with dotted lines and areas containing spindle MTs (squares) are enlarged in the left-hand side panels. Kinetochores are marked with arrowheads. Note that Alp14^NLS^-GFP and Mis12-GBP-mCherry do not completely colocalise; additional Alp14^NLS^-GFP dots are observed in a *dis1∆* cell. This could represent the SPBs, to which Alp14 is normally localised [15,43]. Scale bars, 1 μm (left) and 10 μm (right). (**C**) Spot test. Indicated strains were serially (10-fold) diluted, spotted onto rich YE5S plates containing Phloxine B and incubated at the indicated temperatures for 2–3 d.

**Figure 7 ijms-20-05108-f007:**
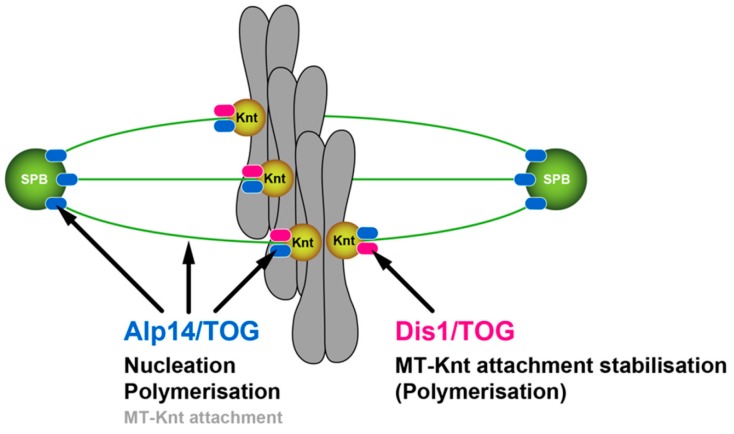
A model of the distinct roles for two XMAP215/TOG proteins in fission yeast. During mitosis, Alp14/TOG, by forming a complex with Alp7/TACC [19], is first recruited to the SPB, where it promotes spindle MT assembly through two different mechanisms: MT nucleation and polymerisation [24]. Alp14 is then transported along spindle MTs towards the kinetochore (Knt), where it binds the Ndc80 kinetochore component [23]. Accordingly, Alp14 is required for MT nucleation, polymerisation and stabilisation. By contrast, Dis1/TOG is recruited to the kinetochore-MT interface through interaction with Ndc80 and Mal3/EB1 without being localised to the SPB [16,20]. Dis1 plays a crucial role in the establishment of stable kinetochore-spindle attachment. These two MAPs have undergone intrinsic functional diversifications, thereby being unreplaceable with each other even if Alp14 and Dis1 are artificially tethered to the kinetochore and the SPB respectively.

**Table 1 ijms-20-05108-t001:** Fission yeast strains used in this study.

Strains	Genotypes	Figures Used	Derivations
MS345	*h^−^ alp14::ura4^+^ leu1 ura4*	1A	Our lab stock
MA003	*h^+^ alp14-GFP-kanR leu1ura4 his2*	1A	Our lab stock
MY2097	*h^−^ alp14-21-GFP-kanR leu1ura4*	1A	This study
TK469	*h^−^ alp14-26-GFP-kanR leu1ura4*	1A	This study
MY2099	*h^−^ alp14-27-GFP-kanR leu1ura4*	1A	This study
TK475	*h^−^ alp14-31-GFP-kanR leu1ura4*	1A	This study
TK484	*h^−^ alp14-32-GFP-kanR leu1ura4*	1A	This study
MY2101	*h^−^ alp14-33-GFP-kanR leu1ura4*	1A	This study
TK358	*h^−^ dis1::hphR alp14-GFP-kanR leu1 ura4*	1A	This study
TK400	*h^−^ alp14-21-GFP-kanR dis1::hphR leu1 ura4*	1A–B	This study
TK405	*h^−^ alp14-26-GFP-kanR dis1::hphR leu1 ura4*	1A–B	This study
TK406	*h^−^ alp14-27-GFP-kanR dis1::hphR leu1 ura4*	1A–B	This study
TK457	*h^−^ alp14-31-GFP-kanR dis1::hphR leu1 ura4*	1A–B	This study
TK458	*h^−^ alp14-32-GFP-kanR dis1::hphR leu1 ura4*	1A–B	This study
TK459	*h^−^ alp14-33-GFP-kanR dis1::hphR leu1 ura4*	1A–B	This study
TK551	*h^+^ plo1-GFP-HA-kanR aur1R-Pnda3-mCherry-atb2 leu1 ura4 his2*	2A–E, 4E	This study
TK583	*h^−^ dis1::ura4^+^ plo1-GFP-HA-kanR aur1R-Pnda3-mCherry-atb2 leu1 ura4*	2A–E, 4E	This study
TK556	*h^−^ alp14-26-myc-hphR plo1-GFP-HA-kanR aur1R-Pnda3-mCherry-atb2 leu1 ura4*	2A–E, 4E	This study
TK585	*h^−^ alp14-26-myc-hphR dis1::ura4^+^ plo1-GFP-HA-kanR aur1R-Pnda3-mCherry-atb2 leu1 ura4*	2A–E, 4D-E	This study
TK586	*h^−^ alp14::ura4^+^ plo1-GFP-HA-kanR aur1R-Pnda3-mCherry-atb2 leu1 ura4*	2A–E, 4E	This study
MO100	*h^−^ cut12-GFP-ura4^+^ aur1R-Pnda3-mCherry-atb2 leu1 ura4*	3A–C, 4A-B	Our lab stock
TK547	*h^+^ dis1::natR cut12-GFP-ura4^+^ aur1R-Pnda3-mCherry-atb2 leu1 ura4 his2*	3A–C	This study
TK580	*h^+^ alp14-26-myc-hphR cut12-GFP-ura4^+^ aur1R-Pnda3-mCherry-atb2 leu1 ura4*	3A–C	This study
TK572	*h^−^ alp14-26-myc-hphR dis1::natR cut12-GFP-ura4^+^ aur1R-Pnda3-mCherry-atb2 leu1 ura4*	3A–C, 4A–C	This study
TK582	*h^+^ alp14::hphR cut12-GFP-ura4^+^ aur1R-Pnda3-mCherry-atb2 leu1 ura4 his2*	3A–C	This study
TK576	*h^−^ dis1-GFP-kanR hphR-GBP-mCherry-alp4 leu1 ura4*	5B–C	This study
TK591	*h^−^ alp14::ura4^+^ dis1-GFP-kanR hphR-GBP-mCherry-alp4 leu1 ura4*	5B–C	This study
TK590	*h^+^ alp7::ura4^+^ dis1-GFP-kanR hphR-GBP-mCherry-alp4 leu1 ura4 his2*	5B–C	This study
TK578	*h^−^ dis1-GFP-kanR leu1 ura4*	5C	This study
TK602	*h^+^ alp14::ura4^+^ dis1-GFP-kanR leu1 ura4 his2*	5C	This study
MS187	*h^−^ alp7::ura4^+^ dis1-GFP-kanR leu1 ura4*	5C	Our lab stock
MY2021	*h^−^ alp14-NLS-GFP-kanR mis12-GBP-6HIS-mCherry-natR leu1 ura4*	6B–C	This study
MY2041	*h^−^ dis1::hphR alp14-NLS-GFP-kanR mis12-GBP-6HIS-mCherry-natR leu1 ura4*	6B–C	This study
MY2027	*h^−^ alp7::ura4^+^ alp14-NLS-GFP-kanR mis12-GBP-6HIS-mCherry-natR leu1 ura4*	6B–C	This study
MS836	*h^+^ alp14-NLS-GFP-kanR leu1 ura4 his2*	6C	Our lab stock
TK360	*h^−^ dis1::hphR alp14-NLS-GFP-kanR leu1 ura4*	6C	This study
MS838	*h^−^ alp7::ura4^+^ alp14-NLS-GFP-kanR leu1 ura4*	6C	Our lab stock

Strains were developed for this study unless otherwise specified. *his2* = *his2-245*; *leu1* = *leu1-32*; *ura4* = *ura4-D18*.

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
