# Peer review of "Two XMAP215/TOG Microtubule Polymerases, Alp14 and Dis1, Play Non-Exchangeable, Distinct Roles in Microtubule Organisation in Fission Yeast"

_ijms, 2019, doi:10.3390/ijms20205108_

Round 1
Reviewer 1 Report
The potentially redundant roles of the two TOG family proteins Alp14 and Dis1 is tested in fission yeast. Temperature-sensitive alleles of alp14 are tested in the absence of Dis1. They provoke an increased mitotic index, with aberrant short spindles, defective pole separation, low microtubule density, spindle collapse, and “cut” phenotype. Changing the respective pole or kinetochore-specific localization of Alp14 and Dis1, by forced targeting of Dis1 to poles or targeting of Alp14 to kinetochores, fails to compensate the loss of the other TOG paralogue. The authors conclude therefore that Alp14 and Dis1 are functionally different and play non-redundant roles in fission yeast.
Although the exact differences that distinguish Alp14 and Dis1 remain to be determined, the information in this manuscript should be valuable for cell biologists, in particular for specialists working on spindle formation in fission yeast. The experiments are carefully performed and the manuscript is well written.
Author Response
Our response:
Thank you very much for this referee’s positive, encouraging comments. He/She has not requested any specific points.
Reviewer 2 Report
The manuscript entitled “ Two XMAP215/TOG microtubule polymerases, Alp14 and Dis1, play non-exchangeable, distinct roles in microtubule organization in fission yeast” from Dr. Takashi Toda group presents data showing that two yeast XMAP215/TOG related proteins, Dis1 and Alp14, although collaborate to ensure the growth and stability of spindle microtubules, cannot substitute for each other.
This is a nicely written manuscript with clearly and logically presented experimental data, elegant figures and interesting discussion. Generally, I liked it very much.
However, I have some sense of insufficiency in case of the last sets of experiments where Authors investigate if Alp14 and Dis1 are functionally exchangeable. Authors showed that Dis1-GFP can be targeted to the SPB and GFP-Alp14 can be targeted to the kinetochore but forcedly recruited proteins were not able to functionally substitute a deleted paralog. The Alp14 interacts with Alp7/TACC while Dis1 interacts with Mal/EB1. Alp7 and Alp14 colocalize to the spindle pole body (SPB) and Alp14 is dislocalized from the SPB and microtubules in alp7 mutants (Sato et al., 2004).
A question is, can Dis1 interact with Alp7 ? Can Alp14 interact with Mal3/EB1? Is this known?
The minor suggestion: the IF images showing merged localization could be larger (Figs 3A, 3C, 5B, 6B).
In my opinion, the obtained data are scientifically sound and of interest to the reader of the IJMS journal.
Author Response
Our responses
Thank you very much for this referee’s positive, thoughtful comments.
Regarding interaction between Dis1 and Alp7 or between Alp14 and EB1/Mal3, in wild-type cells, we have obtained only negative results, namely no interaction, under conditions where we did detect interaction between Dis1 and EB1/Mal3 or between Alp14 and Alp7, respectively. In strains containing the protein-targetting system, we have not performed interaction studies. But we think that this is a surplus experiment that would be checked in the future.
As for image presentations, as suggested by this referee, we now present enlarged images of merged localisation in Figs 3A, 3C, 5B and 6B.